

# 1 LAND-SE: a software for landslide statistically-based
# 2 susceptibility zonation, Version 1.0

**4 Mauro Rossi and Paola Reichenbach**

CNR IRPI, via Madonna Alta 126, 06128 Perugia, Italia
Correspondence to: Mauro Rossi (mauro.rossi@irpi.cnr.it)
**Abstract**
Landslide susceptibility (LS) provides an estimate of the landslide spatial occurrence based on
local terrain conditions. LS has been evaluated in many locations around the world since the
early '80 using distinct modelling approaches, diverse combination of variables, and different
partition of the territory (mapping units). Among the different methods, statistical models have
been largely used to assess LS and several model types have been proposed in the literature. A
recent literature review revealed that authors not always present a complete and comprehensive
assessment of the LS that includes model performance analysis, prediction skills evaluation and
estimation of the errors and uncertainty.
The aim of this paper is to describe LAND-SE (LANDslide Susceptibility Evaluation), software
that performs susceptibility modelling and zonation using statistical models, quantifies the
model performances and the associated uncertainty. The software is implemented in R, a free
software environment for statistical computing and graphics. This provides users with the
possibility to implement and improve the code with additional models, evaluations tools or
output types. The paper describes the software structure, explains input and output, illustrates
specific applications with maps and graphs. The LAND-SE script is delivered with a basic user
guide and three example datasets.

**Keywords**: Landslides, susceptibility, statistical models, zonation, R





## 1 Introduction


Landslide susceptibility (LS) is the likelihood of a landslide occurring in an area based on local
terrain conditions (Brabb, 1984). In mathematical language, LS quantifies the spatial
probability of landslides occurrence in a mapping unit, not considering the temporal probability
of failure or the magnitude of the expected landslides. Landslide susceptibility has been
evaluated in many locations around the world since the early '80. Authors have evaluated LS
using different partition of the territory as mapping units, diversified combination of
explanatory variables and distinct methods. Methods for the LS evaluation and mapping can be
broadly grouped in: geomorphological mapping, analysis of landslide inventories, heuristic or
index-based methods, statistically based models and geotechnical or physically based models
(Guzzetti et al., 1999). Among the different approaches, the statistical models have been largely
used to assess LS. A recent revision of papers on statistical models (Malamud et al., 2014),
have shown that more than 95 different model types were proposed in the literature. Malamud
co-authors grouped them in 20 classes, with the most frequent corresponding to logistic
regression, neural networks and data overlay. The relevant number of statistical models
described in the literature is probably related to the recent increasing number of commercial
and open source packages for statistical analysis that can combine and integrate geographical
data and/or Open Source GIS (i.e. SAGA GIS, GRASS GIS). The review analysis also revealed
that authors not always present a complete and comprehensive assessment of the models
performance and prediction skills evaluations and estimation of the errors and uncertainty.
Since the large variety of applications of statistical approaches, but the scarcity of model
evaluation and quantification of the errors, we have implemented LAND-SE (LANDslide
Susceptibility Evaluation), software developed to prepare landslide susceptibility models and
zonation at basin and regional scale, with specific functions focused to results evaluation and
uncertainty estimation. The software is implemented in R, a free software environment for
statistical computing and graphics (R Core Team, 2015). This provides users with the
possibility to implement and improve the code with additional models, evaluations tools or
output types.
The paper describes LAND-SE structure, explains input and output, illustrates with maps and
graphs, some applications and provides a basic user guide. It is out of the scope of the
manuscript, the description of the characteristics of each model, the advantage/disadvantage of
the model evaluation parameters and the analysis of the model results. We have introduced a



test area only to show and demonstrate possible potential applications and different output of
LAND-SE.
The manuscript is structured as follows: section 2 describes the software, its modelling
approaches and the main output types; section 3 illustrates the test area and describes some
applications and section 4 formalizes some final remarks. The paper is completed by ancillary
materials containing the software code, a user guide and example datasets.

## 2    Software description

LAND-SE, software for landslide susceptibility modelling and zonation was implemented and
improved with respect to the code proposed by Rossi and co-authors in 2010. The new version
is coded in R (R Core Team, 2015) and it is open source. The software holds on the possibility
to perform and combine different statistical susceptibility modelling, evaluate the results and
estimate the associated uncertainty. As compared to the previous version (Rossi et al., 2010),
the main improvements are related to: i) the possibility to use different cartographic units (pixel-
based vs polygon-based); ii) the capacity to perform different type of validation analyses
(spatial, temporal, random); iii) the ability to evaluate the model prediction skills and
performances using success and prediction rate curves (Chung and Fabbri, 1999; 2003); iv) the
possibility to provide results in standard geographical formats (shapefiles, geotiff); v) an
optimization and stabilization of the modelling algorithms; vi) the possibility to use additional
computational parameters to tune the calculation procedure, for the analysis of large dataset.
This software version presents a relevant computer code restructuring (code refactoring),
allowing the implementation of new single statistical approaches (e.g. support vector machines,
regression tree based approaches) that can be added as new modules, preserving the basic
software structure. The new structure simplifies the maintainability and improvement of the
source code.
Figure 1 shows the logical schema of LAND-SE subdivided in the following five functions:

I.    Data input preparation;

II.    Single susceptibility models and zonation;

III.    Combination of single models using a logistic regression approach;

IV.    Evaluation of single and combined LS models;

88          V.    Estimation of uncertainty of single and combined LS models.



### 2.1 Data input preparation

The input data preparation, follows two steps: i) the choice of the cartographic unit and ii) the
selection of the criteria for the definition of the training and the validation dataset.
LAND-SE is designed to use different cartographic units, reducible to pixels or to polygon-like
subdivisions (e.g. slope units, geomorphological subdivisions, administrative boundaries, etc.).
The input data shall be provided in tabular format where each line represents one mapping unit
with the associated attributes. Since raster data cannot be used directly as input, a preliminary
conversion is required to perform the pixel-based analysis.
To identify and separate the training and the validation dataset, different criteria can be adopted.
Temporal, spatial or random subdivisions can be selected guiding the type of validation
analysis. When the temporal validation is selected, secondary information not used in the model
training must be provided for the area under analysis. Adopting a temporal subdivision
approach, the training and the validation set are composed by the same mapping units and the
analysis is performed using the same explanatory variables but different grouping variable (e.
g. a different landslide inventory map, often more recent than what is used during the training
phase). Differently, in the spatial and random approach, the training and the validation dataset
contain different mapping units, characterized by different grouping and explanatory variables.
The main difference between the spatial and the random validation is the method chosen to
separate the training and the validation dataset: in the first case, the datasets are spatially
different (the two areas can be contiguous or not), in the second the subdivision is performed
using a random selection. For the pixel-based approach, the definition of the training and the
validation dataset can follow the same criteria, but in the literature, the subdivision is commonly
performed using a random selection (Van Den Eeckhaut et al., 2010; Felicísimo et al., 2013;
Petschko et al., 2014).

### 2.2 Single susceptibility models estimation (single susceptibility maps)

LAND-SE uses different supervised multivariate statistical models to evaluate the landslide
spatial probability, identifying and quantifying the relation between dependent and independent
variables. According to previous studies (Carrara et al., 1991; Rossi et al., 2010; Guzzetti et al.,
2006, 2012), dependent variable (or grouping variable) is computed as the absence/presence of
landslides in the mapping units and is usually derived from a landslide inventory. The
independent variables (explanatory variables) are obtained from available thematic information





(morphometry, land cover/use, lithology, etc.). Each model is executed in two phases: a the
training phase, where the model reconstructs the relationships between the dependent and the
independent variables, and a validation phase, where these relationships are verified in different
conditions. LAND-SE calculates landslide susceptibility with the following single models
(Rossi et al., 2010): i) linear discriminant analysis (LDA) (Fisher, 1936; Brown, 1998; Venables
and Ripley, 2002), ii) quadratic discriminant analysis (QDA) (Venables and Ripley, 2002), iii)
logistic regression (LR) (Cox, 1958; Brown, 1998; Venables and Ripley, 2002), and iv) neural
network (NN) modelling (Ripley, 1996; Venables and Ripley, 2002). The logistic regression
model was significantly improved with respect to Rossi et al. (2010), substituting the previous
code based on the "Zelig" package (Owen et al., 2013), with a more stable and performing code
based on the "glm" function, included in the well tested base R implementation (R Core Team,

2015).

**2.3 Combined model using a logistic regression approach (combined**
**susceptibility maps)**
Similarly to the previous version, LAND-SE uses a combination model (CM) based on a
logistic regression approach, where the grouping variable is the presence or absence of
landslides in the mapping units, and the explanatory variables are the forecasts of the selected
single susceptibility models (Rossi et al., 2010). Similarly, to the single logistic regression
model, the original code based on the "Zelig" package was substituted with the "glm" function.
LAND-SE allows to enable or not, the execution of the combined model selecting different
combinations of single models.
**2.4 Susceptibility model evaluation**
In the training phase, LAND-SE reconstructs the relationships between dependent and
independent variables and evaluates the prediction skills of single and combined models (i.e.
the capability to predict the original data). In the validation phase, LAND-SE verifies the results
in different conditions and evaluates the models capability to predict independent data. Models
output of both phases are evaluated using the same tools and in particular the following
statistical metrics and indices:
• The dependence among explanatory variables (Belsley, 1991; Hendrickx, 2012);
• Contingency tables (i.e. confusion matrixes) (Jollifee and Stephenson, 2003);



• Contingency plots or fourfold plots summarizing the mapping units correctly and
incorrectly classified by the models (Jollifee and Stephenson, 2003);
• Error maps showing the geographical distribution of the mapping units correctly and
incorrectly classified by the models (Rossi et al., 2010);
• Plots showing receiver operating characteristic (ROC) curves (Green and Swets, 1966;
Mason and Graham, 2002; Fawcett, 2006) and the associated Area Under Curve (AUC)
statistics;
• Evaluation plots showing the variation of the sensitivity ("hit rate"), the specificity (1-
false positive rate), and of the Cohen's kappa index (Cohen,1960);
• Success and prediction rate curves (Chung and Fabbri, 1999; 2003)
The description and discussion of the characteristics, advantage/drawbacks of these statistical
metrics/indices are out of the scope of the manuscript and they will not be described in detail.
**2.5 Uncertainty evaluation (single and combined susceptibility zonations)**
For each single and combined model, LAND-SE evaluates and quantifies the uncertainty
adopting a "bootstrapping" re-sampling technique (Efron, 1979; Davison and Hinkley, 2006).
In the training phase, a user-specified number of runs are performed varying the selected
dataset. Descriptive statistics for the probability (susceptibility) estimates, including the mean
($\mu$) and the standard deviation ($\sigma$), are obtained from an ensemble of model runs (i.e. a user-
defined number of LAND-SE simulations are executed to obtain the two descriptive statistics).
Such information is portrayed in plots showing estimates for the model uncertainty in each
mapping unit and in maps showing the geographical distribution of the uncertainty (Guzzetti et
al., 2006; Rossi et al., 2010). To model the uncertainty associated to each LS zonation, the mean
and the standard deviation are fitted using a parabolic function (Figure 3D). Such function is
used to estimate the uncertainty in the validation phase. The map showing the geographical
distribution of the uncertainty can provide additional and relevant information for the use of LS
zonation in environmental planning studies. A proper interpretation of the map may provide for
each mapping unit a proxy of a degree of confidence associated to the LS estimate.
**2.6 SW output formats**
LAND-SE can be executed in two different modes: the *standard* that provides textual and
graphical results stored respectively in .txt and .pdf, and the *geomode* providing also



geographical output as shapefiles and GeoTIFF. Some output (i.e the success and prediction
rate curves) are produced only in the *geomode* because they require geographical data
(shapefile) as additional input. A complete list of the output with a detailed description is
provided in the supplementary material (LAND-SE_UserGuide.pdf).

## 3    LAND-SE applications
To show software functionalities and output types, LAND-SE was applied in a test area.
Different configurations were selected to perform the following analysis:
• Polygon-based landslide susceptibility zonation;
• Pixel-based landslide susceptibility zonation;
• Landslide susceptibility scenarios zonation.
The applications use different mapping units and distinct schema to select the training and
validation dataset. One analysis is focused to illustrate the use of LAND-SE to evaluate the
impact of different scenarios of land use on LS. LAND-SE results can be considered relevant
information in environmental planning and management.
### 3.1    Description of the example area and available data
A small area was selected to show applications and output of LAND-SE. The area is located in
the eastern portion of the Briga catchment (Figure 2), in the Messina province (Sicily, South
Italy). It has elevation values ranging from the sea level to about 500 m and terrain gradient in
the range of 0° - 81°. Landslides, including shallow soil slides and debris flows, deep-seated
rotational and translational slides, and complex and compound failures (Varnes, 1984), are
abundant, and caused primarily by rainfall (Ardizzone et al., 2012; Reichenbach et al., 2014;
2015). On 1 October 2009, the Briga catchment and the surrounding areas were hit by an intense
storm (Maugeri and Motta, 2011) that triggered more than 1000 shallow landslides, mainly
shallow soil slides and debris flows (Varnes, 1984), caused 37 fatalities, numerous injured
people and severe damages in the affected villages and along the transportation network.
After the event, a detailed landslide inventory map at 1:10,000 scale was prepared for the entire
Briga catchment (Ardizzone et al., 2012). The inventory was obtained through a combination
of field surveys carried out in the period from October to November 2009, and visual
interpretation of pre-event and post-event stereoscopic and pseudo-stereoscopic aerial





photographs. The inventory map shows the distribution and types of landslides triggered by the
1 October 2009 rainfall event (Figure 2), and the distribution and types of pre-existing
landslides. In addition, two maps reporting the land use in different periods were prepared
exploiting available aerial photographs and Very High Resolution (VHR) satellite imagery
(Reichenbach et al., 2014; 2015). The first map was derived from the analysis of the same black
and white aerial photograph used to map pre-event landslides. The second map was obtained
from the analysis of two QuickBird satellite images taken the first on 2 September 2006 and
the second on 8 October 2009 (Mondini et al., 2011).
In the area, landslide susceptibility zonation were prepared using two mapping units: pixels and
slope-units. The slope-units (SU) are terrain subdivisions bounded by drainage and divide lines
(Carrara et al., 1991). SU were outlined using a 5-meter resolution DEM obtained resampling
the VH resolution DEM provided by the Italian national Department for Civil Protection and
using a recently developed *r.slopeunits* module (Marchesini et al., 2012;-Alvioli et al., 2016).
The size and the geometrical characteristics of the SU are controlled by modeling parameters
defined by the user including the minimum half-basin area (Metz et al., 2011) and the slope
aspect variability. In the study area, the procedure identified 238 SU which represent the
polygon-based mapping units for the determination of LS. To explain the spatial distribution of
landslides (Carrara et al., 1991; 1995), for each slope-unit, we calculated the percentage of the
event landslides as dependent (grouping) variable and the following explanatory variables: i)
descriptive statistics (range, mean, standard deviation) of elevation and slope; ii) the percentage
covered by each land use class; and iii) the percentage covered by old landslides.
For the pixel-based analysis, we used the VH resolution DEM (1m x 1m) that accounts for
about 5 million cells. Maps of the elevation, slope, land use and of the presence/absence of old
landslides, were used as explanatory variables in the analysis. The presence/absence of event
landslides was used as dependent variable (Carrara et al., 1991, 1995; Guzzetti et al., 2006).
The data originally in polygon format were first converted in raster and all the data were
converted to the tabular format to be suited for LAND-SE (see LAND-SE_UserGuide.pdf for
details).





### 3.2 Polygon-based landslide susceptibility zonation


This example is focused to illustrate landslide susceptibility zonation prepared using the slope-
unit as mapping unit. Two spatial criteria were used to define the training and validation dataset,
the first based on a random selection and the second on the subdivision of the entire catchment
in two contiguous areas (Nord and South).
In the first case, the training set contained 70% of the total slope-units and the validation
corresponded to the entire basin. Landslide susceptibility models were trained using a subset of
available data and results were applied in validation to the entire study area. Figure 3 shows the
main graphical and geographical outputs obtained during the training and the validation phases,
including susceptibility, error and uncertainty maps, fourfold (contingency) plot, success and
prediction rate curves, ROC plot, evaluation and uncertainty plots. For simplicity, the figure
shows only results of the combined model, but outputs for each single model are available and
can be exploited for further analysis. In the example, the random selection criteria resulted in
similar training and validation performances (Figure 3). This application simulates LS zonation
for large territory, where landslides information is spotted and do not cover the entire study
area. In such conditions, training cannot be performed on the entire area and a random selection
of the training dataset, within the surveyed area, is a reasonable solution.
In the second case, the SU located in the Northern part of the Briga catchment with respect to
the main river, were used as training set and the SU located in the Southern portion as validation
set. Figure 4 shows outputs, including susceptibility maps for the combined model, success and
prediction rate curves, and ROC plots. As shown in Figure 4, the spatial subdivision resulted in
good model skill analysis, but reduced validation performances, underlying a poor spatial
exportability of the model (i.e. poor applicability of the resulting model coefficients to different
study areas). This type of application simulates LS zonation for areas where landslides
information required to train the model, is available only for a portion of the area. Results
obtained in the training phase are then applied to estimate susceptibility to the portion of the
territory where landslide data are not available. This application can be useful to evaluate the
possibility to use the same model output in different portion of territory or in different areas.

### 3.3 Pixel-based landslide susceptibility zonation


This example shows a landslide susceptibility zonation prepared using the pixel as mapping
unit. A random selection was chosen to prepare the training set and the validation was



performed applying results on the entire study area. For the purpose, in the training set all the
pixels corresponding to landslides and an equal number of pixels in stable areas were selected.
Figure 5 shows the main outputs of the combined model prepared for the entire area during the
validation phase, including susceptibility, error and uncertainty maps, fourfold (contingency)
plot, prediction rate curve, ROC plot, evaluation and uncertainty plots.
This example simulates a common and widespread susceptibility zonation approach that
exploits pixel-based analysis at basin and regional scale. In such conditions, reasonable
calculation times with a limited loss of performances can be reached  to training the model with
a random selected subset and applying results to the entire study area. As shown in Figure 5,
although the training was performed with a subset of the data, the model performance for the
entire study area is adequate and acceptable.

### 3.4  Landslide susceptibility scenarios zonation

This example illustrates how LAND-SE can be utilized to evaluate the impact of different land-
use scenarios on landslide susceptibility zonation (Reichenbach et al. 2014, 2015). The current,
the past and possible future land-use distributions were evaluated on landslide susceptibility
classes. Single models (linear discriminant analysis, quadratic discriminant analysis and logistic
regression) and a combined model were prepared, exploiting the 2009 event landslides as
grouping variable and morphological and land-use classes as explanatory variables.
To evaluate the influence of land use change on landslide susceptibility zonation, results
obtained with the 2009 land use map were applied using the 1945 land use distribution. Figure
6 portrays on the left, the combined model prepared using the 2009 land use map, and on the
right the zonation obtained applying the results to the 1954 land use cover. Moreover, to
estimate the effect of land use distribution, we have designed different scenarios obtained
changing the 2009 land use cover. Assuming an increase in the forested areas, we have
considered three types of changes computed at the slope unit scale resulting in the following
scenarios: i) 75% decrease in the pasture extent (Scenario 1); ii) 75% reduction of both pasture
and cultivated areas (Scenario 2); and iii) 75% decrease in bare soil where the slope-unit mean
angle was greater than 15° together with 75% decrease in pasture areas (Scenario 3). A fourth
scenario was prepared assuming the effect of a forest fire in the south-west part of the area,
where we simulated a reduction of the forested cover and an increase in bare soil (Scenario 4).



For each scenario, figure 7 shows the CM zonation and the success rate curve measuring the
fitting performance of each model.
Analyses of the scenarios confirm how land use changes significantly affect the spatial
distribution of unstable/stable slopes. This information can be used to evaluate the
consequences of land use change on vulnerability and risk. Moreover, the proposed approach
can be helpful to analyse the potential effects of land use planning and management on slope
instability.

**4   Final remarks**
A recent review analysis on landslide statistical models revealed a large variety of statistical
types, but a significant scarcity of a complete and comprehensive evaluation of the models
performance and prediction skills (Malamud et al. 2014). Moreover assessment of the input
data quality (Ardizzone et al. 2002), discussion on the scale applicability and the quantification
of errors and uncertainty associate to the models are limited. In the recent years there has been
an increase number of commercial and open source packages for statistical analysis that
integrate geographical data and/or Open Source GIS, but software dedicated to landslide
susceptibility zonation using statistical models is not available.
LAND-SE is an open source SW that performs LS modelling, zonation, results evaluation and
associated uncertainty estimation using graphs, map and statistical metrics filling the lacks of
the large variety of statistical methods already available. We think further improvements may
include additional models (i.e. forest tree analysis), tools for the input data preparation, tools
for the visualization of results available now only in textual format (i.e. test of the collinearity
evaluation, number of significant variables). Moreover, the software can be applied and
customized to different applications, providing the users with the possibility to implement and
improve the code with additional models, evaluations tools or output types. LAND-SE can also
be used to prepare models to predict particular types of slope movements (e.g. debris flow
source areas, Carrara et al., 2008) or can be customized to evaluate the probability of spatial
occurrence of completely diversified natural phenomena.

**Acknowledgements**



The implementation and improvement of LAND-SE with respect to the version published by
Rossi et al. (2010), was supported by the FP7 LAMPRE Project (Landslide Modelling and
Tools for vulnerability assessment preparedness and recovery management, EC contract n.

31238).


**Code availability and licence**
The LAND-SE code is provided as supplementary materials together with:
1. the software user guide (LAND-SE_UserGuide_v1_03mar2016.docx);
2. datasets containing the software script (LAND-SE_v30_20160118.R), the configuration

files (LAND-SE_configuration_spatial_data.txt, LAND-SE_configuration.txt) and input

files (training.txt, training.shp, validation.txt, validation.shp) relative to three examples

applications: (i) polygon-based landslide susceptibility zonation with a random selection

of the training dataset and a validation on a larger area; (ii) polygon-based landslide

susceptibility zonation with training and validation performed in two different contiguous

areas; (iii) pixel-based landslide susceptibility zonation with a random selection of the

training dataset and a validation on a larger area.

LAND-SE Copyright (C) Mauro Rossi. LAND-SE is free software; it can be redistributed or
modified under the terms of the GNU General Public (either version 2 of the License, or any
later version) as published by the Free Software Foundation. The program is distributed in the
hope that it will be useful, but WITHOUT ANY WARRANTY; without even the implied
warranty of  MERCHANTABILITY or FITNESS FOR A PARTICULAR PURPOSE. See the
GNU General Public License for more details.

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




**Figures**

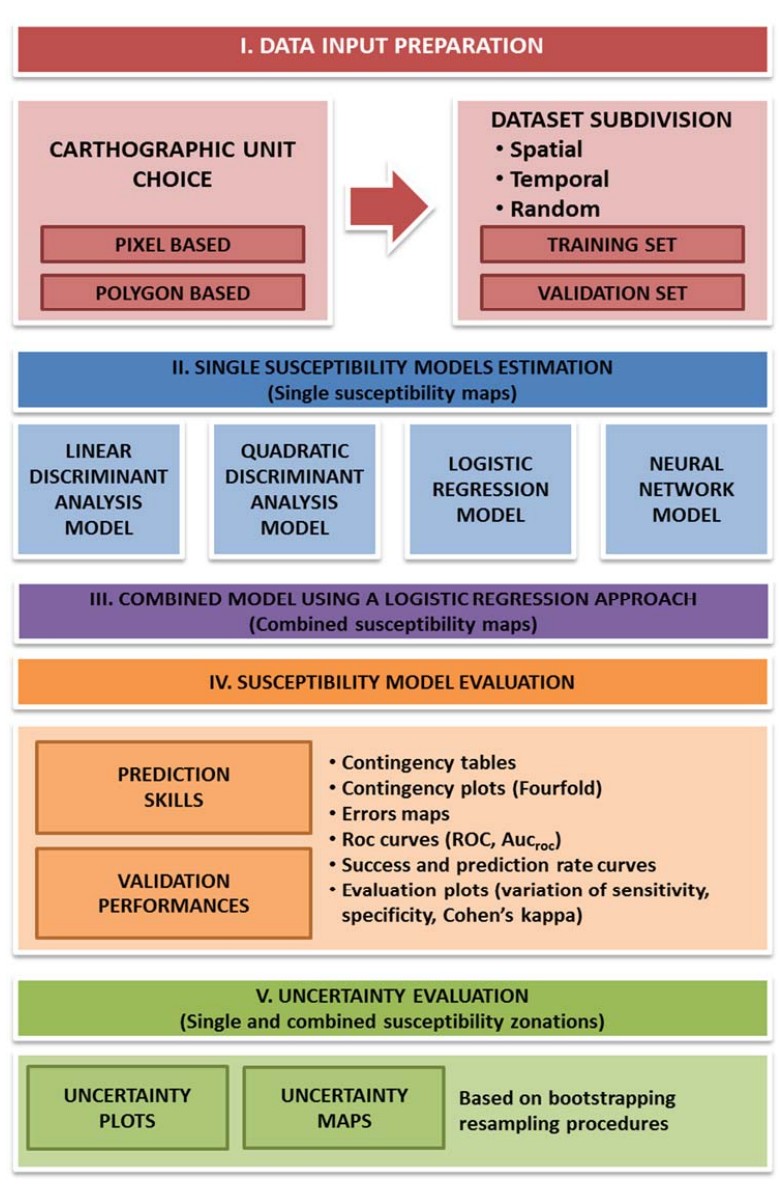



Figure 1. Logical schema of the LAND-SE software for landslide susceptibility modelling and
zonation.



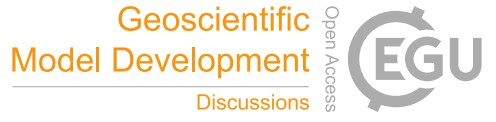

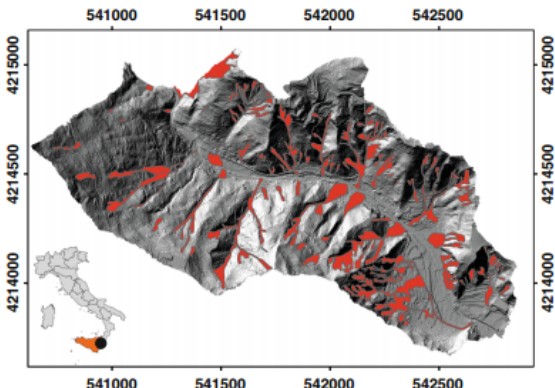



Figure 2. Shaded relief of the study area located in the Briga catchment, along the Ionian coast
of Sicily (Italy). Red polygons show landslides triggered by the October 1, 2009 rainfall event.



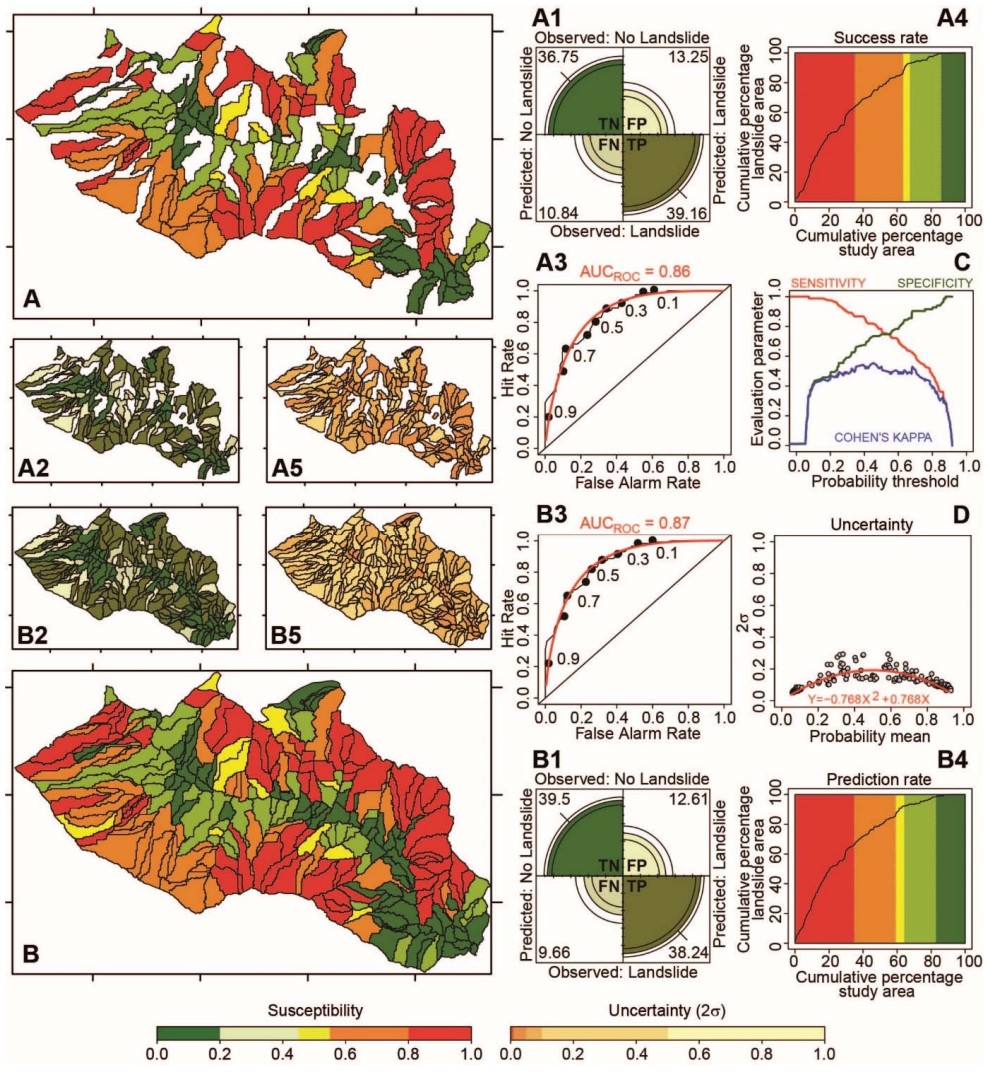

Figure 3. Landslide susceptibility maps (CM) for the training dataset (A) and the validation dataset (B) classified in five unequally spaced classes (see legend). (A1, B1) fourfold plots summarizing the number of true positives (TP), true negatives (TN), false positives (FP), and false negatives (FN); (A2, B2) maps of the distribution of the four categories of slope units reported in the fourfold plots; (A3, B3) ROC plots; (A4, B4) success and prediction rate curves; (C) variation in the model sensitivity, specificity, and Cohen's kappa index; (D) plot showing measures of the model error ($2\sigma$) vs. the mean probability ($\mu$), for each slope unit, (black circle); (A5, B5) maps of the geographical distribution of the model error.





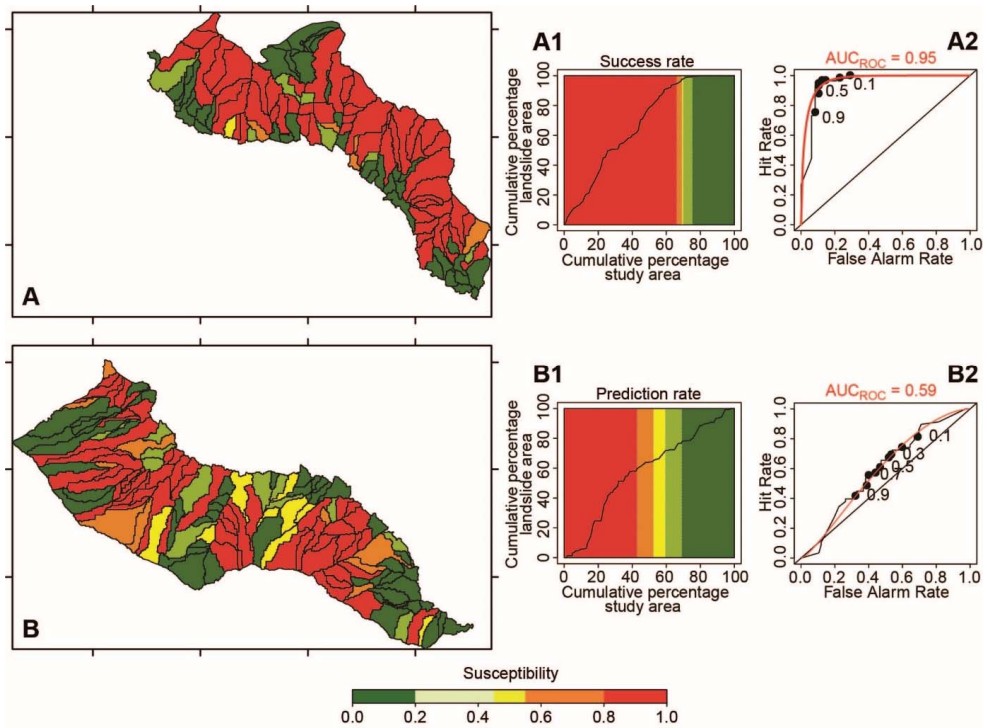



Figure 4. Landslide susceptibility maps (CM) for the training dataset (A: Northern part) and the
validation dataset (B: Southern part) of the test area, classified in five unequally spaced classes
(see legend). (A1, B1) success and prediction rate curves; (A2, B2) ROC plots.



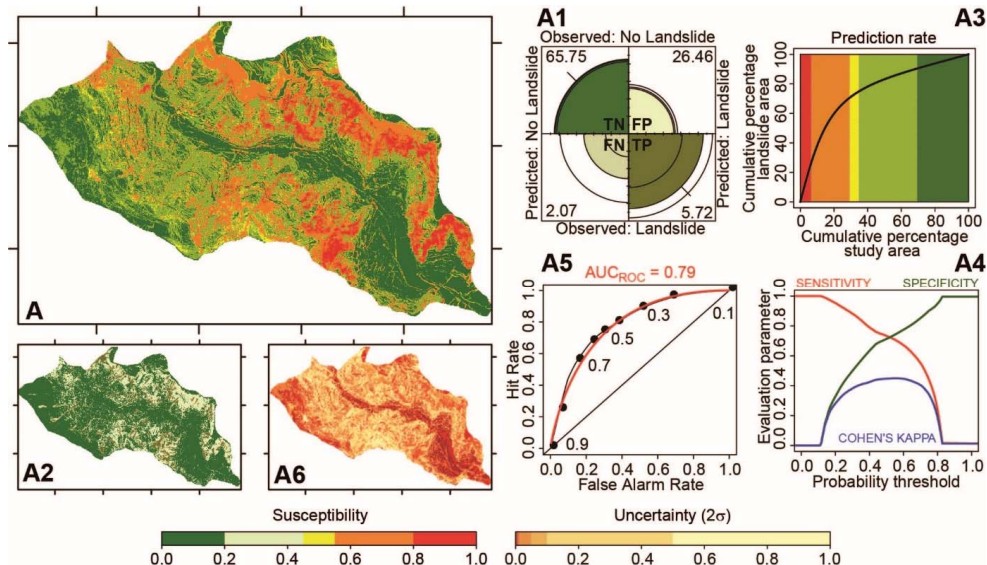



Figure 5. Pixel-based landslide susceptibility map (CM) of the test area (A) classified in five
unequally spaced classes (see legend). (A1) fourfold plot summarizing the number of true
positives (TP), true negatives (TN), false positives (FP), and false negatives (FN); (A2) map of
the distribution of the four categories reported in the fourfold plot; (A3) prediction rate curve;
(A4) variation in the model sensitivity, specificity, and Cohen's kappa index; (A5) ROC plot;
(A6) map of the geographical distribution of the model error.






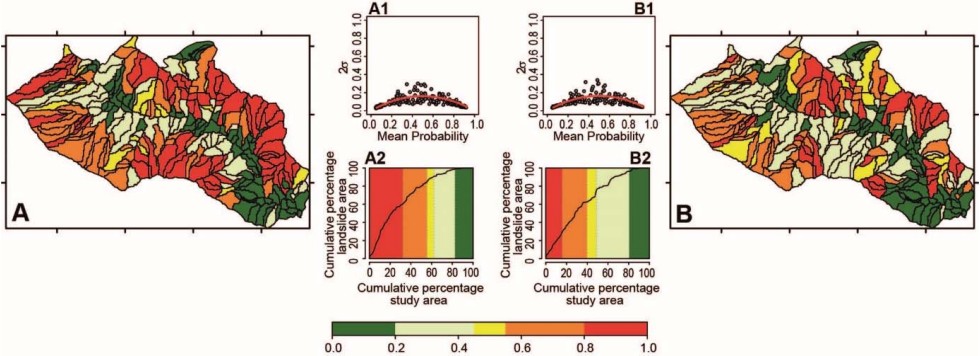

Figure 6. (A) Landslide susceptibility map (CM) prepared using the 2009 land use and (B) using the 1954 land use cover. LS maps are classified in five unequally spaced classes (see legend); (A1, B1) plot showing the model uncertainty estimated in each slope unit; (A2, B2) success rate curves.






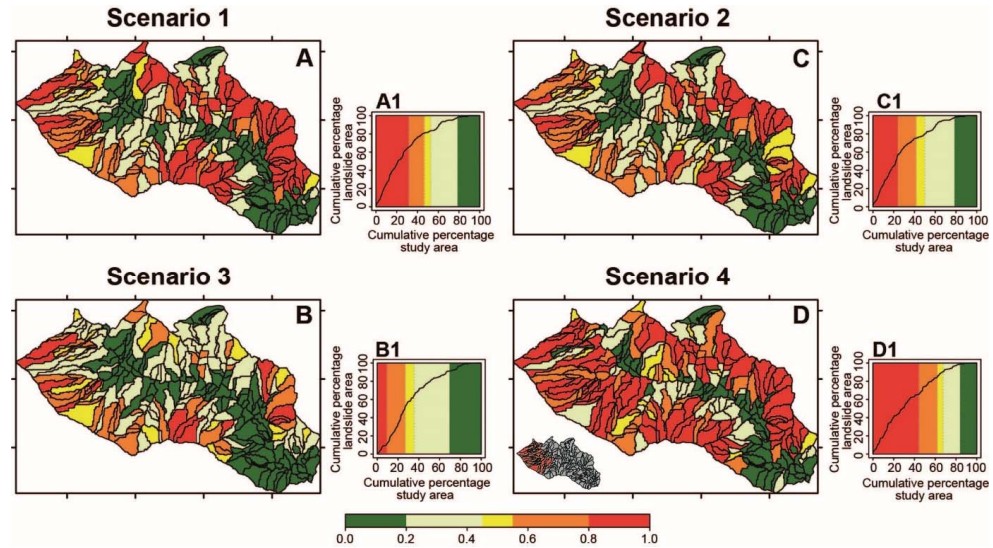



Figure 7. (A, B, C, D) Landslide susceptibility maps (CM) classified in five unequally spaced
classes prepared using different land use scenario; (A1, B1, C1, D1) success rate curves.