# Peer review of "LAND-SE: a software for statistically-based landslide susceptibility zonation, Version 1.0"

_Geoscientific Model Development, 2016_

## Referee Comment (RC1) · S. Nasiri (Referee) · 1 Jun 2016

The paper is well structured and quite innovative in integrating various statistical analysis and converting landslide technical terms and physical parameters through computer programing into landslide susceptibility assessment software. The authors defined the Landslide Susceptibility terminology well and took advantage of their comprehensive literature review to address those parameters which are influential for landslide occurrence. However there are some minor errors and technical corrections especially in the figures which need to be revised in their paper:

In the Abstract, Page1-Line9: While the author is going to introduce "Landslide susceptibility" with the abbreviation of "LS", it is recommended to be more specific about the

topic by using "Assessment or Evaluation" terms like: Susceptibility (LS) assessment provides. . ." or ". . .Landslide Susceptibility (LS) evaluation provides. . ."

In the Abstract, Page1-Line9: The definition of "Landslide susceptibility": ". . . an estimate of the landslide spatial occurrence based on. . ." it is better to change to ". . . a relative estimation of the landslide spatial occurrence based on. . ."

In the Abstract, Page1-Line11 to 13: Due to the repetition of these two sentences in the beginning of the introduction, this should be paraphrased in the Abstract

Figure 2: there is some simple cartography revision which is needed for this map including adding a legend and a scale, and showing different types of landslides base on the Varnes-1984 classification.

Figure 6 & Figure 7; at the bottom scale bar, labeling of "Susceptibility" is missed for the susceptibility scale bar.
* * *

---

## Referee Comment (RC2) · Anonymous Referee #2 · 4 Jul 2016

Review

**LAND-SE: a software for landslide statistically-based susceptibility zonation, Version 1.0**

Mauro Rossi and Paola Reichenbach

Submitted to Geoscientific Model Development

**General comments**

The authors of this manuscript are presenting a very interesting software, or R-script, which allows the user to perform landslide susceptibility modeling and a detailed assessment of the quality of the model in terms of model performance and standard error of the model output. This can be done by a variety of graphs and maps in a tabular and spatial visualization, automatically generated by the script. While most of the presented script was published before by Rossi et al. (2010), the authors clearly indicate the alterations and optimizations that have been performed on this original script since the year 2010. The possibility to include new models (like regression trees) is a clear step forward and novel to the old version from 2010. Furthermore, they implemented the more stable "glm" function for the logistic regression modeling, which is widely used in statistical landslide susceptibility modeling and in ecological modeling.

Landslide susceptibility modeling is performed worldwide more and more often to provide local communities with spatial information on where the occurrence of a landslide is more probable and where people have to take precautions when outlining new housing areas or when building new houses. The authors correctly state, that in many cases the susceptibility modeling is done very simply and often the limitations of the models themselves are not reported. However, the range of model performance and the general error of the prediction as presented by the standard deviation of the modeled probability is often not reported on. With this tool practitioners, or rather fellow scientists, can perform landslide susceptibility modeling and have a detailed look on model performance and uncertainties. However, as far as I understood from the manuscript, the effect of repeatedly drawing different samples for training and testing the model is only considered for the model uncertainty, but not for the effect on the model performance measures as reported on by other authors in the field. Please see the specific comments for more details on that.

The presentation of the manuscript and software is sound, however some minor English spelling and Grammar errors were identified which make the language sometimes less fluent or precise. Furthermore, the manuscript can still be improved by adding some more details on the methods and assumptions included in the software (e.g. on the sampling and partitioning into training and test sample procedure). Although the authors state clearly that e.g. the discussion of the advantages and disadvantages of the one or the other model performance measure was beyond the scope of the paper, a general discussion of the limitations of the models or the presented software and its results is missing and should be included as this is also demanded in the guidelines of this journal. Additionally, the audience of the software is unclear as the amount of information on limitations and proper usage might vary significantly if the software is aimed to be used by informed, modeling experienced scientists or less modeling experienced practitioners (please see the more detailed thoughts in the specific comments section).

Given these general comments and the following specific comments I would like to suggest minor to major revisions for this scientifically valuable manuscript.

**Specific comments**

I would like to suggest some restructuring of the manuscript as some essential information, such as how the sampling of presence data was performed for grid cells or terrain units, is only presented in the applications section. This is crucial information on the model which should be presented with the model in section 2. Please consider including more detail on this in the input data preparation section. E.g. at lines 109-112: Please clarify how exactly a landslide is represented in the model. In a later section it says that the entire landslide polygon represented in pixel of the proper resolution is included in the modeling. Given that in literature this is treated with different option I was wondering if this the only option in LAND-SE or if the user can choose how the landslides should be represented. Other authors such as Atkinson et al. (1998), Atkinson and Massari (2011) and Van Den Eeckhaut et al. (2006) report this step with different sampling designs in their research. Please clarify why you chose to include the entire polygon. A valuable source for discussing this might be the rather recent paper of Regmi et al. (2014) regarding the effects of which information on the landslides is included as presence data, on the modeling results, maybe for the discussion of the limitations of the software. Regmi et al. (2014) found that it makes a big difference which information of the landslide is used for the modeling, therefore it might also affect the models which are based on pixel as terrain units in this manuscript. Please consider including this in the discussion of the limitations of the model results.

Another area of interest is the sampling of training and test data. Here the question arises from reading the manuscript if the sampling of training and test data was done within the R-script or if this is something the user has to prepare beforehand? From the User Manual I read that the user has to perform that subsampling before. However, with that the repeatability of the model is at risk. Is there an option to include this in the model (e.g. similarly to the bootstrapping?)? Please provide some more details on this in the manuscript.

While the very often in literature suggested and very advanced possibility to create training and test samples randomly, spatially and temporally is implemented in the software, it is striking, that the sampling is only performed once for the model fitting and evaluation. I understand from reading the manuscript, that with the bootstrapping only the uncertainty or variation of the model in terms of the mean predicted probability and its standard deviation was assessed by testing multiple models. However, it is unclear how often the model should be run at a minimum or maximum to achieve reliable results and why this was not used to compute the range in the AUROC values and other model performance estimates as well. In my understanding, using the bootstrapping to compute repeated spatial or random training and test subsamples and therefore multiple performance measures, would be the same as repeated spatial or non-spatial cross validation as often mentioned in recent literature. This does not seem to portray the state of the art in this field as recently multiple authors have performed repeated random or spatial subsampling for assessing the model performance in terms of AUROC, spatial transferability and thematic consistency and have shown that with the sample, the performance measures change distinctly (e.g. Goetz et al., 2015; Heckmann et al., 2014; Petschko et al., 2014; von Ruette et al., 2011; Steger et al., 2016). I would like to suggest to include this into the model or address why it was not done. Furthermore, the word uncertainty is used rather generally. Please make sure to be specific which type of uncertainty (model form uncertainty, uncertainty from the input data, etc.) is analyzed.

If I understood correctly the examples for the landslide susceptibility modeling are originating from two published studies from Reichenbach et al. 2014 and 2015. However, for most of chapter 3 this

stays rather unclear. Please be more out front about this fact if my assumption applies by referring to the studies at the beginning of the chapter.

The modeling of the landslide susceptibility scenarios depending on the land cover changes is very interesting. However, it stays unclear for the reader how the land cover scenarios were computed (e.g. a regression model like CLUE-V or other ways). Please consider inserting some information on this here by referring to the original study it was performed in.

The final remarks are very similar to the abstract and particularly to the introduction of the submitted manuscript. I would like to suggest to rewrite this section to give a more critical view or discussion on the software, its limitations and proper scale of application.

Throughout the manuscript I was wondering who the target audience for the software is. Please specify that somewhere in the manuscript. Depending on the target audience I was wondering if the user manual could contain some help on how to interpret the results of this software (e.g. the value range of an AUROC value and its meaning for the model performance, or the susceptibility classes). While reading I was also wondering if the landslide susceptibility classes are provided by the software, or if this is something that the user can choose. You see from my questions that I am getting very excited about the software as it could help many users worldwide to enhance their understanding for the local landslide susceptibility. Therefore, I would be very happy if you could address some of my questions in the manuscript. The supplementary material is well prepared and will aid any user of any level to run the susceptibility model.

**Technical corrections**

**Text**

The title seems appropriate for the paper. However, given the fact, that the submitted software is an optimized version of a script published by the authors in 2010, I wonder if it should be given a different version name (e.g. 1.1 or 2.0). Furthermore, I suggest a change in the sentence structure of the title to: LAND-SE: a software for statistically-based landslide susceptibility zonation, Version X.Y. This structure change seems more logical to me from an English Grammar point of view. However, I am no English native speaker, which is why I would like to suggest a thorough proof read of the entire manuscript by an English native speaker. In this section I will indicate some spelling and grammar errors as far as I noticed them. But this list might not be complete or correct! In the following I will give the sentence as it was in the manuscript with underlined parts that I would propose to add or change in the text.

Line 21: "… with additional models, evaluation tools or output types." Please delete the "s" at evaluation.

Line 22: "…, explains input and output and illustrates specific applications with maps and graphs." Please consider including the "and".

Line 32: "… since the early 1980." Maybe including the "19" would make sense to be more accurate.

Line 33: "… using different partitioning of the territory as mapping units, analysis of landslide inventories,…" Please consider changing from "partition" to "partitioning".

Line 39: "Malamud and his co-authors grouped them in 20 classes,…". Please include "and his".

Line 41: "According to them the relevant number of statistical models…" Please consider including "According to them".

Line 45,46: Please consider changing this part to: "… comprehensive assessment of the model performance, the prediction skill evaluations and…"

Line 49: "Susceptibility Evaluation), a software developed to prepare…" Please consider inserting the "a".

Line 50: ",… with specific functions focused on result evaluation…" Please consider changing accordingly.

Line 53: ", evaluation tools or output types." Please consider changing accordingly.

Line 55: ", explains input and output, illustrates them with maps and graphs…" Please consider changing accordingly.

Lines 56 to 58: Please rephrase this sentence as it is very difficult to understand.

Line 59: "… test area to demonstrate the range of applications and different outputs of LAND-SE." Please consider shortening and simplifying the sentence accordingly.

Line 63: Please exchange "ancillary" for "supplemental"

Line 66: "LAND-SE, a software …" Please consider changing accordingly.

Line 69: "… and combine different statistical susceptibility modelling methods, evaluate …" Please consider changing accordingly.

Line 77: "datasets" instead of "dataset" Please consider changing accordingly.

Line 84 and 89: "Input data preparation" instead of "data input preparation" Please consider changing accordingly.

Line 145-146: "Model outputs" instead of "Models output" Please consider changing accordingly.

Line 177: Please consider spelling out the word software here and in any future occasion instead of using the abbreviation SW as it is easier to read and understand for the reader.

Line 198: "The elevation ranges from the sea level to about 500m and the terrain gradient ranges from 0° to 81°". Please consider the changes in this sentence for the manuscript.

Line 218: "…, landslide susceptibility zonation was prepared …" Please consider changing accordingly.

Line 222: Please include the software it was written for. I assume GRASS GIS? Or a GRASS GIS tool within QGIS?

Line 252-253: "This application simulates LS zonation for a large territory, where landslide information is spotted and does not cover the …" Please consider changing accordingly.

Line 261: Please consider using the word "transferability" instead of "exportability" as used by von Ruette et al. 2011 and Petschko et al. 2014.

Line 262-263: Please use the singular: "landslide information"

Line 277: Please rephrase this sentence to be more specific on which loss of performance is commented on there.

Line 288: Please rephrase the sentence to be more specific on which results you are referring to here. I assume the resulting equation to describe the statistical relationship (e.g. the regression equation with intercept and coefficients)?

**Figures**

The figures are generally done beautifully and are very informative. I have only one minor remark. Please consider including a sentence of reference at the figure caption of figure one regarding the scale and geographical location of the study area. Figure 1 is the only figure that includes the coordinates around the map box. Please prepare the reader that this information is eliminated in the following figures but always stays the same for all figures. Additionally, a small figure included in Figure 1 showing the location of the study area within Italy or Sicily would be of high interest.

**References**

Atkinson, P., Jiskoot, H., Massari, R. and Murray, T.: Generalized linear modelling in geomorphology, Earth Surf. Process. Landf., 23(13), 1185–1195, 1998.

Atkinson, P. M. and Massari, R.: Autologistic modelling of susceptibility to landsliding in the central apennines, Italy, Geomorphology, In Press, Accepted Manuscript, doi:doi: DOI: 10.1016/j.geomorph.2011.02.001, 2011.

Goetz, J. N., Brenning, A., Petschko, H. and Leopold, P.: Evaluating machine learning and statistical prediction techniques for landslide susceptibility modeling, Comput. Geosci., 81, 1–11, doi:10.1016/j.cageo.2015.04.007, 2015.

Heckmann, T., Gegg, K., Gegg, A. and Becht, M.: Sample size matters: investigating the effect of sample size on a logistic regression susceptibility model for debris flows, Nat. Hazards Earth Syst. Sci., 14(2), 259–278, doi:10.5194/nhess-14-259-2014, 2014.

Petschko, H., Brenning, A., Bell, R., Goetz, J. and Glade, T.: Assessing the quality of landslide susceptibility maps – case study Lower Austria, Nat Hazards Earth Syst Sci, 14(1), 95–118, doi:10.5194/nhess-14-95-2014, 2014.

Regmi, N. R., Giardino, J. R., McDonald, E. V. and Vitek, J. D.: A comparison of logistic regression-based models of susceptibility to landslides in western Colorado, USA, Landslides, 11(2), 247–262, doi:10.1007/s10346-012-0380-2, 2014.

von Ruette, J., Papritz, A., Lehmann, P., Rickli, C. and Or, D.: Spatial statistical modeling of shallow landslides—Validating predictions for different landslide inventories and rainfall events, Geomorphology, 133(1–2), 11–22, doi:10.1016/j.geomorph.2011.06.010, 2011.

Steger, S., Brenning, A., Bell, R., Petschko, H. and Glade, T.: Exploring discrepancies between quantitative validation results and the geomorphic plausibility of statistical landslide susceptibility maps, Geomorphology, 262, 8–23, doi:10.1016/j.geomorph.2016.03.015, 2016.

Van Den Eeckhaut, M., Vanwalleghem, T., Poesen, J., Govers, G., Verstraeten, G. and Vandekerckhove, L.: Prediction of landslide susceptibility using rare events logistic regression: a case-study in the Flemish Ardennes (Belgium), Geomorphology, 76(3–4), 392–410, 2006.

---

## Referee Comment (RC3) · Anonymous Referee #3 · 14 Jul 2016

In this manuscript, the authors introduce a new model for assessing landslide susceptibility. I consider that there are needs for that kind of model for a variety of objectives. So, I consider that there is the worthy for publication in GMD after the moderate revision. I commented several points to clarify the advantage of their proposed model.

Section 2.2 I cannot fully understand about single susceptibility models in LAND-SE. In Introduction, the authors introduced the recent review by Malamud et al. (2014) and they argued that more than 95 different models were proposed and can be grouped into 20 classes. Also, they bited a significant scarcity of a complete and comprehensive evaluation of the models performance and prediction skills in Final Remarks. I believe that the authors tried to overcome these problems, but it was unclear. I suggest that the

authors have to describe "single susceptibility models" in LAND-SE in more detail. To clarify the advantage of LAND-SE for complete and comprehensive evaluation of the models performance, I think that the authors have to show their answer to the following questions: 1. How many models in 95 models did LAND-SE cover? 2. How many Malamud's groups did LAND-SE cover? Perhaps if the authors summarized single susceptibility models in LAND-SE into a table, it should be helpful for readers.

Section 2.3 I cannot understand the method to combine results of single susceptibility models from the manuscript. Although the method was already presented in the previous paper of Rossi et al. (2010), I think that the authors have to show the method of combination. If the method us totally the same as the method presented by Rossi et al. (2010), the authors have to clarify it.

Section 2.5 I consider that this section is one of key parts of this study, since the authors noted that the quantification of errors and uncertainty of the models are limited (L312-L313). However, the review and description of the method for quantifying errors and uncertainty are not adequate. So, the authors have to show detailed information about the method. Also, the authors have to review the uncertainty analysis and show the reason why the authors chose "bootstrapping".

Section 3.4 I think that the authors did not validate the results of this section using the data. They just calcurated the landslide susceptibility based on the scenarios of landuse. If the authors want to show one of examples of possibility of LAND-SE, I can understand meaning of this section, but I cannot agree with the last paragraph in this section (L302-306). If the authors want to note the effectiveness of LAND-SE for testing effects of landuse change on landslide susceptibility, they have to validate their calculation results.

---

## Author Comment (AC1) · 10 Aug 2016

M. Rossi and P. Reichenbach

mauro.rossi@irpi.cnr.it

We have revised the article "LAND-SE: a software for landslide statistically-based susceptibility zonation, Version 1.0", by Mauro Rossi and Paola Reichenbach, following the reviewers comments. The revised version includes corrections and modifications to the main text, figures and reference list. We have also revised the software user guide included in the supplementary material. We have not changed the software and data examples. In the attached zip file you'll find: 1) the letter with answers to editor and referees; 2) the revised manuscript with figures; 3) the revised user guide. Regards Mauro Rossi

[Figure]

Please also note the supplement to this comment:
http://www.geosci-model-dev-discuss.net/gmd-2016-60/gmd-2016-60-AC1-
supplement.zip

———————————————————

---

## Author Response (AR1)

**Consiglio Nazionale delle Ricerche**
**Istituto di Ricerca per la Protezione Idrogeologica**

To the Editor of
"Geoscientific Model Development"

Perugia, 10 August 2016

Subject: Submission of the revised version of the article "*LAND-SE: a software for landslide statistically-based susceptibility zonation, Version 1.0*"

Dear Editor,

This letter is attached to the submission of the revised version of the article "*LAND-SE: a software for landslide statistically-based susceptibility zonation, Version 1.0*", by Mauro Rossi and Paola Reichenbach. The revised version includes corrections and modifications to the main text, figures and reference list. We have also revised the software user guide included in the supplementary material. We have not changed the software and data examples.

We would like to thank the editor and the reviewers for their constructive comments and suggestions that improved the comprehension and the quality of the article. As suggested by Reviewer #2, we changed the article title to "*LAND-SE: a software for statistically-based landslide susceptibility zonation, Version 1.0*".

In the following, you will find point by point responses to comments and suggestions of each reviewer. Reviewers' comments are in italic and our answers and changes in regular fonts.

**Responses to Reviewer #1 S. Nasiri**

Comments

*The paper is well structured and quite innovative in integrating various statistical analysis and converting landslide technical terms and physical parameters through computer programing into landslide susceptibility assessment software. The authors defined the Landslide Susceptibility terminology well and took advantage of their comprehensive literature review to address those parameters which are influential for landslide occurrence.*

*However there are some minor errors and technical corrections especially in the figures which need to be revised in their paper:*

*In the Abstract, Page1-Line9: While the author is going to introduce "Landslide susceptibility" with the abbreviation of "LS", it is recommended to be more specific about the topic by using "Assessment or Evaluation" terms like: Susceptibility (LS) assessment provides. . ." or ". . .Landslide Susceptibility (LS) evaluation provides. . ."*

Mauro Rossi • IRPI CNR • via Madonna Alta 126 • 06128 Perugia • Italy
Tel. +39 075 5014.421 • Fax. +39 075 5014.420 • http://www.irpi.cnr.it/ • e-mail: mauro.rossi@irpi.cnr.it

We agree with the suggestion and modified accordingly.

*In the Abstract, Page1-Line9: The definition of "Landslide susceptibility": ". . . an estimate of the landslide spatial occurrence based on. . ." it is better to change to ". . . a relative estimation of the landslide spatial occurrence based on. . ."*
We agree with the suggestion and modified the text in "... a relative estimate of the landslide spatial occurrence based on..."

*In the Abstract, Page1-Line11 to 13: Due to the repetition of these two sentences in the beginning of the introduction, this should be paraphrased in the Abstract*
We agree and we modified the text from line 11 as follows:
"A literature review revealed that LS evaluation has been performed in many study areas worldwide using different methods, model types, different partition of the territory (mapping units) and a large variety of geo-environmental data. Among the different methods, statistical models have been largely used to evaluate LS, but the minority of articles presents a complete and comprehensive LS assessment that includes model performance analysis, prediction skills evaluation and estimation of the errors and uncertainty."

*Figure 2: there is some simple cartography revision which is needed for this map including adding a legend and a scale, and showing different types of landslides base on the Varnes-1984 classification.*
We agree and we modified the figure following the reviewer's suggestions.

*Figure 6 & Figure 7; at the bottom scale bar, labeling of "Susceptibility" is missed for the susceptibility scale bar.*
We agree and we have modified the two figures adding in the captions of Figures 3, 4, 5, 6, 7 the sentence: "Maps coordinates and scale bar are shown in Figure 2."

Responses to Reviewer #2

*General comments*
*The authors of this manuscript are presenting a very interesting software, or R-script, which allows the user to perform landslide susceptibility modeling and a detailed assessment of the quality of the model in terms of model performance and standard error of the model output. This can be done by a variety of graphs and maps in a tabular and spatial visualization, automatically generated by the script. While most of the presented script was published before by Rossi et al. (2010), the authors clearly indicate the alterations and optimizations that have been performed on this original script since the year 2010.*

*The possibility to include new models (like regression trees) is a clear step forward and novel to the old version from 2010. Furthermore, they implemented the more stable "glm" function for the logistic regression modeling, which is widely used in statistical landslide susceptibility modeling and in ecological modeling.*

*Landslide susceptibility modeling is performed worldwide more and more often to provide local communities with spatial information on where the occurrence of a landslide is more probable and where people have to take precautions when outlining new housing areas or when building new houses. The authors correctly state, that in many cases the susceptibility modeling is done very simply and often the limitations of the models themselves are not reported. However, the range of model performance and the general error of the prediction as presented by the standard deviation of the modeled probability is often not reported on. With this tool practitioners, or rather fellow scientists, can perform landslide susceptibility modeling and have a detailed look on model performance and uncertainties. However, as far as I understood from the manuscript, the effect of repeatedly drawing different samples for training and testing the model is only considered for the model uncertainty, but not for the effect on the model performance measures as reported on by other authors in the field.*

We thank the reviewer to have highlighted this interesting issue. As correctly stated by the reviewer, the sampling procedure included in LAND-SE is focused to estimate the uncertainty associated to the susceptibility zonation prepared using a given landslide and environmental dataset. Actually, the script provides an estimate of the performance variability in the training and validation phases providing confidence levels in the ROC plots and in the contingency (fourfold) plots.

The sensitivity of the susceptibility models associated with different sampling strategies is not included in the script. The analyses of model performances, model sensitivity and of variables significance can be set up in different ways and commonly require different approach to subdivide training and validation sets. Sample sizes and ratio between the training and validation sizes can be decided following different sampling strategies that can be decided by the expert who is running the model. Given the numerous possibility of variations required to set this type of analysis, we decided to do not include such functionalities in the current LAND-SE release, but we designed and implemented a command line interface (see §S5 of the LAND-SE User Guide V 1.0) to make this analysis possible using external procedures. The user can select the size and type of training and validation sets outside the LAND-SE script and then run the code for each samples evaluating and comparing the variability of the results. We have tested this approach for different purposes and found it convenient allowing flexibility in deciding sampling strategies but also in using specific language/scripting environment. The user can implement a script using any language/scripting environment (e.g. Pyhton, GRASS, ArcGIS) finalized to prepare different data set/samples and then run LAND-SE using the command line interface. As additional information, we are currently developing a scripting tool coded in R for the data preparation which will include functionalities for the aforementioned specific purpose. The tool is still in preliminary and prototype version, but will be shortly available upon request.

To make the above explicit we added at the end of section **2.5 Uncertainty evaluation (single and combined susceptibility zonations)** the following text:

"The sampling procedure implemented in LAND-SE is only focused to the estimation of the uncertainty associated to the susceptibility zonation. However, the software also outputs estimates of

the performance variability in the training and validation phases providing confidence levels in the ROC plots (NCAR, 2014) and in the fourfold or contingency plots (Meyer et al., 2015). In addition, the execution of analyses that investigate sensitivity or variability of model results when varying inputs (e.g. using sampling procedures), is facilitated by the LAND-SE command line interface, that makes these analyses possible using external procedures."

We also add two additional references:

- NCAR - Research Applications Laboratory: verification: Weather Forecast  Verification Utilities, R package version 1.41. http://CRAN.R-project.org/package=verification, 2014.
- Meyer, D., Zeileis, A., Hornik, K.: vcd: Visualizing Categorical Data, R package version 1.4-0, 2015.

*Please see the specific comments for more details on that.*
*The presentation of the manuscript and software is sound, however some minor English spelling and Grammar errors were identified which make the language sometimes less fluent or precise. Furthermore, the manuscript can still be improved by adding some more details on the methods and assumptions included in the software (e.g. on the sampling and partitioning into training and test sample procedure). Although the authors state clearly that e.g. the discussion of the advantages and disadvantages of the one or the other model performance measure was beyond the scope of the paper, a general discussion of the limitations of the models or the presented software and its results is missing and should be included as this is also demanded in the guidelines of this journal.*
*Additionally, the audience of the software is unclear as the amount of information on limitations and proper usage might vary significantly if the software is aimed to be used by informed, modeling experienced scientists or less modeling experienced practitioners (please see the more detailed thoughts in the specific comments section).*

We believe challenging and interesting a detailed discussion on the limitations of the classification models in terms of advantages and disadvantages of each statistical approach applied for the susceptibility estimation. However, we thing this requires a detailed statistically-driven discussion that is beyond the scope of the paper. To make clear LAND-SE limitations and the target software audience, we modified the manuscript and the software user guide, as follows.

We added the following text at the end of section **4. Final Remarks** of the main manuscript:
"LAND-SE is mainly designed to evaluate landslide susceptibility from basin (medium) to regional scale (small to very small scale).The quality and significant of model outputs is highly related to the scale, accuracy and resolution of landslide and environmental input data. In the field of landslide susceptibility zonation, LAND-SE is designed to be properly and productively used by experienced geomorphologists. Experienced practitioners are expected to use the code, with the support of experts in the field of environmental planning and management for a correct and reliable interpretation and exploitation of the results. A proper LAND-SE execution requires: (i) a basic knowledge of R language to run the script; (ii) experience on multivariate statistical models and on their evaluation skills/metrics (ROC plot, contingency table and plots, success/prediction rate curves, etc.); (iii) GIS skills to prepare and handle input data; and (iv) specific expertise for a correct and reliable interpretations of the results.

All the modelling types implemented in LAND-SE are basically statistical classification techniques applicable to any multivariate analysis with a binary grouping (dependent or response) variable. This makes the code flexible and appropriate to other scientific fields and usable, with minor customization and tailoring, by user with different expertise."

We added section **S1. User skills** in the LAND-SE User Guide with the following text:
"LAND-SE (LANDslide Susceptibility Evaluation) is a software developed to prepare landslide susceptibility models and zonation at basin and regional scale, with specific functions focused on results evaluation and uncertainty estimation. The software is implemented in R, a free software environment for statistical computing and graphics (R Core Team, 2015). In the field of landslide susceptibility zonation, LAND-SE is designed to be properly and productively used by experienced geomorphologists. Experienced practitioners are expected to use the code, with the support of experts in the field of environmental planning and management for a correct and reliable interpretation and exploitation of the results.
A proper LAND-SE execution requires: (i) a basic knowledge of R language to run the script; (ii) experience on multivariate statistical models and on their evaluation skills/metrics (ROC plot, contingency table and plots, success/prediction rate curves, etc.); (iii) GIS skills to prepare and handle input data; and (iv) specific expertise for a correct and reliable interpretations of the results.
All the modelling types implemented in LAND-SE are basically statistical classification techniques applicable to any multivariate analysis with a binary grouping (dependent or response) variable. This makes the code flexible and appropriate to other scientific fields and usable, with minor customization and tailoring, by user with different expertise."

We added the following text at the end of section **S3. Input and data specifications** in the LAND-SE User Guide:
"LAND-SE is highly demanding in terms of RAM, mainly for the pixel-based approach. The demand of RAM depends on: i) the size of the study area and the pixel resolution; ii) the number of explanatory variables; and iii) the number and type of model selected. When LAND-SE is applied to very large areas, calculations may require very long computational time. A significant improvement in the script execution could be obtained using fast CPUs. A more efficient and advanced solution, that might be improved in the future, consider a code parallelization."

*Given these general comments and the following specific comments I would like to suggest minor to major revisions for this scientifically valuable manuscript.*

*Specific comments*
*I would like to suggest some restructuring of the manuscript as some essential information, such as how the sampling of presence data was performed for grid cells or terrain units, is only presented in the applications section. This is crucial information on the model which should be presented with the model in section 2. Please consider including more detail on this in the input data preparation section. E.g. at lines 109-112: Please clarify how exactly a landslide is represented in the model. In a later section it says that the entire landslide polygon represented in pixel of the proper resolution is included*

*in the modeling. Given that in literature this is treated with different option I was wondering if this the only option in LAND-SE or if the user can choose how the landslides should be represented. Other authors such as Atkinson et al. (1998), Atkinson and Massari (2011) and Van Den Eeckhaut et al. (2006) report this step with different sampling designs in their research. Please clarify why you chose to include the entire polygon. A valuable source for discussing this might be the rather recent paper of Regmi et al. (2014) regarding the effects of which information on the landslides is included as presence data, on the modeling results, maybe for the discussion of the limitations of the software. Regmi et al. (2014) found that it makes a big difference which information of the landslide is used for the modeling, therefore it might also affect the models which are based on pixel as terrain units in this manuscript. Please consider including this in the discussion of the limitations of the model results.*

Thanks for this comment. We are aware of this issue and we have recently published a research on the effect of different landslide sampling strategies in a grid-based bi-variate statistical susceptibility model. (Hussin et al., 2016)

In section **2.1 Data input preparation** of the main manuscript we added:

"The choice of the mapping unit is crucial because it also determines how landslides are sampled to prepare the training and prediction (validation) subsets for the susceptibility modelling. In grid-based susceptibility assessments, several strategies are used to sample landslide pixels, the more frequent are: (1) single pixel selected as the centroid of the entire landslide or the scarp area; (2) all the pixels within the entire landslide body or the scarp area; (3) the main scarp upper edge (MSUE) approach which selects pixels on and around the landslide crown-line; and (4) the seed-cell approach that selects pixels within a buffer polygon around the upper landslide scarp area and sometimes part of the flanks of the accumulation zone (Atkinson et al.,1998; Atkinson and Massari, 2011; Goetz et al., 2015; Heckmann et al., 2014; Hussin et al., 2016; Regmi et al., 2014; Van Den Eeckhaut et al., 2006). The analysis of model sensitivity to different landslide mapping strategies and the significance of different variables combinations can be performed using LAND-SE preparing different input files. Given the numerous possibility of variations required to set this type of evaluation, we decided not to include such functionalities in the current LAND-SE release, but we designed and implemented a command line interface (see §S5 of the LAND-SE User Guide V 1.0) to make this analysis possible using external procedures."

*Another area of interest is the sampling of training and test data. Here the question arises from reading the manuscript if the sampling of training and test data was done within the R-script or if this is something the user has to prepare beforehand? From the User Manual I read that the user has to perform that subsampling before. However, with that the repeatability of the model is at risk. Is there an option to include this in the model (e.g. similarly to the bootstrapping?)? Please provide some more details on this in the manuscript.*
*While the very often in literature suggested and very advanced possibility to create training and test samples randomly, spatially and temporally is implemented in the software, it is striking, that the sampling is only performed once for the model fitting and evaluation. I understand from reading the manuscript, that with the bootstrapping only the uncertainty or variation of the model in terms of the mean predicted probability and its standard deviation was assessed by testing multiple models. However, it is unclear how often the model should be run at a minimum or maximum to achieve reliable*

*results and why this was not used to compute the range in the AUROC values and other model performance estimates as well. In my understanding, using the bootstrapping to compute repeated spatial or random training and test subsamples and therefore multiple performance measures, would be the same as repeated spatial or non-spatial cross validation as often mentioned in recent literature. This does not seem to portray the state of the art in this field as recently multiple authors have performed repeated random or spatial subsampling for assessing the model performance in terms of AUROC, spatial transferability and thematic consistency and have shown that with the sample, the performance measures change distinctly (e.g. Goetz et al., 2015; Heckmann et al., 2014; Petschko et al., 2014; von Ruette et al., 2011; Steger et al., 2016). I would like to suggest to include this into the model or address why it was not done. Furthermore, the word uncertainty is used rather generally. Please make sure to be specific which type of uncertainty (model form uncertainty, uncertainty from the input data, etc.) is analyzed.*

Thanks for these comments; we have added additional text in the manuscript to answer to your suggestions. See other answers for details.

*If I understood correctly the examples for the landslide susceptibility modeling are originating from two published studies from Reichenbach et al. 2014 and 2015. However, for most of chapter 3 this stays rather unclear. Please be more out front about this fact if my assumption applies by referring to the studies at the beginning of the chapter.*

Thanks for the comment. We modified the beginning of section **3. LAND-SE applications** as follows:
"To show LAND-SE functionalities and output types, we use as example the landslide susceptibility modelling and zonation originating from two articles published by Reichenbach and co-authors (2014; 2015). In the area selected as example, we perform the following analysis, using different configurations:"

*The modeling of the landslide susceptibility scenarios depending on the land cover changes is very interesting. However, it stays unclear for the reader how the land cover scenarios were computed (e.g. a regression model like CLUE-V or other ways). Please consider inserting some information on this here by referring to the original study it was performed in.*

Thanks for the comment. We specified in the text (section **3.4 Landslide susceptibility scenarios zonation**) that we have designed different scenarios using a heuristic and empirical approach.
The option to use a dynamic spatially explicit, land use and land cover change model (like CLUE-V) is a very interesting suggestion.

*The final remarks are very similar to the abstract and particularly to the introduction of the submitted manuscript. I would like to suggest to rewrite this section to give a more critical view or discussion on the software, its limitations and proper scale of application.*

*Throughout the manuscript I was wondering who the target audience for the software is. Please specify that somewhere in the manuscript. Depending on the target audience I was wondering if the user manual could contain some help on how to interpret the results of this software (e.g. the value range of an AUROC value and its meaning for the model performance, or the susceptibility classes).*

Thanks for these comments. See previous answers for details on the changes done in the manuscript.

*While reading I was also wondering if the landslide susceptibility classes are provided by the software, or if this is something that the user can choose.*

Landslide susceptibility classes are provided by the software, but an expert user can change them in the script. We added the following text in section **S4. List of outputs** of the user guide.
"The number and width of landslide susceptibility classes used to prepare maps and histograms can be modified by the user. This can be done changing the default values of the following variables in the script:

- breaks.histogram.values<-c(0,0.2,0.45,0.55,0.8,1)

- breaks.map.susceptibility<-c(0,0.2,0.45,0.55,0.8,1.0001)"

*You see from my questions that I am getting very excited about the software as it could help many users worldwide to enhance their understanding for the local landslide susceptibility. Therefore, I would be very happy if you could address some of my questions in the manuscript. The supplementary material is well prepared and will aid any user of any level to run the susceptibility model.*

*Technical corrections*
*Text*
*The title seems appropriate for the paper. However, given the fact, that the submitted software is an optimized version of a script published by the authors in 2010, I wonder if it should be given a different version name (e.g. 1.1 or 2.0). Furthermore, I suggest a change in the sentence structure of the title to: LAND-SE: a software for statistically-based landslide susceptibility zonation, Version X.Y. This structure change seems more logical to me from an English Grammar point of view.*
We have used version 1.0 since this is the first time we select a name for the software. The script published with the article paper in 2010 was without a name.

*However, I am no English native speaker, which is why I would like to suggest a thorough proof read of the entire manuscript by an English native speaker. In this section I will indicate some spelling and grammar errors as far as I noticed them. But this list might not be complete or correct! In the following I will give the sentence as it was in the manuscript with underlined parts that I would propose to add or change in the text.*

*Line 21: "... with additional models, evaluation tools or output types." Please delete the "s" at evaluation.*
Done

*Line 22: "..., explains input and output and illustrates specific applications with maps and graphs." Please consider including the "and".*
Done

*Line 32: "... since the early 1980." Maybe including the "19" would make sense to be more accurate.*
Done

*Line 33: "... using different partitioning of the territory as mapping units, analysis of landslide inventories,..." Please consider changing from "partition" to "partitioning".*
Done

*Line 39: "Malamud and his co-authors grouped them in 20 classes,...". Please include "and his".*
Done

*Line 41: "According to them the relevant number of statistical models…" Please consider including "According to them".*
Done

*Line 45,46: Please consider changing this part to: "... comprehensive assessment of the model performance, the prediction skill evaluations and…"*
Done

*Line 49: "Susceptibility Evaluation), a software developed to prepare…" Please consider inserting the "a".*
Done

*Line 50: ",… with specific functions focused on result evaluation…" Please consider changing accordingly.*
Done

*Line 53: ", evaluation tools or output types." Please consider changing accordingly.*
Done

*Line 55: ", explains input and output, illustrates them with maps and graphs…" Please consider changing accordingly.*
Done

*Lines 56 to 58: Please rephrase this sentence as it is very difficult to understand.*
Done

*Line 59: "... test area to demonstrate the range of applications and different outputs of LAND-SE."*
*Please consider shortening and simplifying the sentence accordingly.*
Done

*Line 63: Please exchange "ancillary" for "supplemental"*
Done

*Line 66: "LAND-SE, a software ..." Please consider changing accordingly.*
Done

*Line 69: "... and combine different statistical susceptibility modelling methods, evaluate ..." Please*
*consider changing accordingly.*
Done

*Line 77: "datasets" instead of "dataset" Please consider changing accordingly.*
Done

*Line 84 and 89: "Input data preparation" instead of "data input preparation" Please consider*
*changing accordingly.*
Done

*Line 145-146: "Model outputs" instead of "Models output" Please consider changing accordingly.*
Done

*Line 177: Please consider spelling out the word software here and in any future occasion instead of*
*using the abbreviation SW as it is easier to read and understand for the reader.*
Done

*Line 198: "The elevation ranges from the sea level to about 500m and the terrain gradient ranges from*
*0° to 81°". Please consider the changes in this sentence for the manuscript.*
Done

*Line 218: "..., landslide susceptibility zonation was prepared ..." Please consider changing*
*accordingly.*
Done

*Line 222: Please include the software it was written for. I assume GRASS GIS? Or a GRASS GIS tool*
*within QGIS?*
Done

*Line 252-253: "This application simulates LS zonation for a large territory, where landslide*
*information is spotted and does not cover the ..." Please consider changing accordingly.*

Done

*Line 261: Please consider using the word "transferability" instead of "exportability" as used by von Ruette et al. 2011 and Petschko et al. 2014.*
Done

*Line 262-263: Please use the singular: "landslide information"*
Done

*Line 277: Please rephrase this sentence to be more specific on which loss of performance is commented on there.*
We changed the sentence as follows:
"In such conditions, reasonable calculation times can be reached training the model with a random selected subset and applying results to the entire study area. Dealing with large dataset, we experienced that training the models using reduced samples (randomly selected) affects slightly the susceptibility model results and performances with a minor increase in the model uncertainty."

*Line 288: Please rephrase the sentence to be more specific on which results you are referring to here. I assume the resulting equation to describe the statistical relationship (e.g. the regression equation with intercept and coefficients)?*
Section **3.4 Landslide susceptibility scenarios zonation** has been changed adding comments on the analysis of the results. In particular:
1) "This example illustrates how LAND-SE can be utilized to evaluate the impact of different land-use scenarios on landslide susceptibility zonation (Reichenbach et al. 2014, 2015) comparing the distribution of stable/unstable slope units and the success rate curves."
2) "Zonation maps obtained with the same models but using the 1954 land use map show a significant reduction in the number of unstable SU. Success rate curves reveal a decrease in the model fitting performance when using the 1954 land use map, due to a reduction of slope units classified as unstable and an increase in stable terrain. In particular, the expansion of bare soil to the detriment of forested areas in the 56 years from 1954 to 2009, determined a general increase in the susceptibility."
3) "The qualitative comparisons of the maps and of the success rate curves obtained for the different scenarios confirm how land use changes significantly affect the spatial distribution of unstable/stable slopes (Reichenbach et al., 2014)."

*Figures*
*The figures are generally done beautifully and are very informative. I have only one minor remark. Please consider including a sentence of reference at the figure caption of figure one regarding the scale and geographical location of the study area. Figure 1 is the only figure that includes the coordinates around the map box. Please prepare the reader that this information is eliminated in the following figures but always stays the same for all figures.*

We agree and we added in the captions of Figure 3,4,5,6,7 the sentence: "Maps coordinates and scale bar are shown in Figure 2."

*Additionally, a small figure included in Figure 1 showing the location of the study area within Italy or Sicily would be of high interest.*
Due to the very small size of the study area compared to the Sicily regional boundary, we decided to indicate the location of the basin with a point, as in the original figure.

*References*
1. *Atkinson, P., Jiskoot, H., Massari, R. and Murray, T.: Generalized linear modelling in geomorphology, Earth Surf. Process. Landf., 23(13), 1185–1195, 1998.*
2. *Atkinson, P. M. and Massari, R.: Autologistic modelling of susceptibility to landsliding in the central apennines, Italy, Geomorphology, In Press, Accepted Manuscript, doi:doi: DOI: 10.1016/j.geomorph.2011.02.001, 2011.*
3. *Goetz, J. N., Brenning, A., Petschko, H. and Leopold, P.: Evaluating machine learning and statistical prediction techniques for landslide susceptibility modeling, Comput. Geosci., 81, 1– 11, doi:10.1016/j.cageo.2015.04.007, 2015.*
4. *Heckmann, T., Gegg, K., Gegg, A. and Becht, M.: Sample size matters: investigating the effect of sample size on a logistic regression susceptibility model for debris flows, Nat. Hazards Earth Syst. Sci., 14(2), 259–278, doi:10.5194/nhess-14-259-2014, 2014.*
5. *Petschko, H., Brenning, A., Bell, R., Goetz, J. and Glade, T.: Assessing the quality of landslide susceptibility maps – case study Lower Austria, Nat Hazards Earth Syst Sci, 14(1), 95–118, doi:10.5194/nhess-14-95-2014, 2014.*
6. *Regmi, N. R., Giardino, J. R., McDonald, E. V. and Vitek, J. D.: A comparison of logistic regression-based models of susceptibility to landslides in western Colorado, USA, Landslides, 11(2), 247–262, doi:10.1007/s10346-012-0380-2, 2014.*
7. *von Ruette, J., Papritz, A., Lehmann, P., Rickli, C. and Or, D.: Spatial statistical modeling of shallow landslides - Validating predictions for different landslide inventories and rainfall events, Geomorphology, 133: 1–2, 11-22, 2011.*

We added all the references suggested by the referee.

<hr>

Responses to Anonymous Referee #3

<hr>

*In this manuscript, the authors introduce a new model for assessing landslide susceptibility. I consider that there are needs for that kind of model for a variety of objectives. So, I consider that there is the worthy for publication in GMD after the moderate revision. I commented several points to clarify the advantage of their proposed model.*
*Section 2.2 I cannot fully understand about single susceptibility models in LAND-SE. In Introduction, the authors introduced the recent review by Malamud et al. (2014) and they argued that more than 95 different models were proposed and can be grouped into 20 classes. Also, they bited a significant scarcity of a complete and comprehensive evaluation of the models performance and prediction skills in*

*Final Remarks. I believe that the authors tried to overcome these problems, but it was unclear. I suggest that the authors have to describe "single susceptibility models" in LAND-SE in more detail. To clarify the advantage of LAND-SE for complete and comprehensive evaluation of the models performance, I think that the authors have to show their answer to the following questions:*

*1. How many models in 95 models did LAND-SE cover?*
As described in **Section 2.2**, LAND-SE evaluates landslide susceptibility using four single models: i) linear discriminant analysis (LDA), ii) quadratic discriminant analysis (QDA), iii) logistic regression (LR), and iv) neural network (NN) modelling. All the four model types are used by other authors as revealed in the articles analysed by Malamud et al. (2014). Logistic regression and neural networks are two of the model types more frequently used in the literature.

*2. How many Malamud's groups did LAND-SE cover?*
Both Rossi M. and Reichenbach P. are co-authors of the article "Malamud, B., Mihir, M., Reichenbach, P., and Rossi, M.: D6.3-Report on standards for landslide susceptibility modelling and terrain zonations, LAMPRE FP7 Project deliverables, http://www.lampre-project.eu, 2014."

*Perhaps if the authors summarized single susceptibility models in LAND-SE into a table, it should be helpful for readers.*
The four single models are listed in section **2.2 Single susceptibility models estimation (single susceptibility maps)** of the main text; in the table **S1** and they also are listed in the caption of tables **S1** and **S4** of the LAND-SE User Guide**.** We do not believe an additional table would add more information.

*Section 2.3 I cannot understand the method to combine results of single susceptibility models from the manuscript. Although the method was already presented in the previous paper of Rossi et al. (2010), I think that the authors have to show the method of combination. If the method us totally the same as the method presented by Rossi et al. (2010), the authors have to clarify it.*
In section **2.3 Combined model using a logistic regression approach (combined susceptibility maps)** we describe the combined model prepared using a logistic regression model explaining the grouping/explanatory variables. We modify the first paragraph of the section **2.3** as follows:
"LAND-SE uses a combination model (CM) based on a logistic regression approach, where the grouping variable is the presence or absence of landslides in the mapping units, and the explanatory variables are the forecasts of the selected single susceptibility models (Rossi et al., 2010). Similarly, to the single logistic regression model, the original code based on the "Zelig" package was substituted with the "glm" function. LAND-SE allows to enable or not, the execution of the combined model selecting different combinations of single models."

*Section 2.5 I consider that this section is one of key parts of this study, since the authors noted that the quantification of errors and uncertainty of the models are limited (L312-L313). However, the review and description of the method for quantifying errors and uncertainty are not adequate. So, the authors have to show detailed information about the method. Also, the authors have to review the uncertainty analysis and show the reason why the authors chose "bootstrapping".*

We added the following text:

"For each single and combined model, LAND-SE evaluates and quantifies the uncertainty adopting a "bootstrapping" approach. Bootstrapping is a resampling technique for estimating the distributions of statistics based on independent observation. Bootstrapping can refer to any test or metric that relies on random sampling with replacement (Efron, 1979; Davison and Hinkley, 2006). The technique has been largely used to estimate errors and uncertainties associated to classification models (among the others, Kuhn and Kjell, 2013)."

A new reference was added:

Kuhn, M., and Kjell, J.: Applied predictive modeling. New York: Springer, 2013.

Additional changes to section **2.5 Uncertainty evaluation (single and combined susceptibility zonations)** are illustrated in the answers to Reviewer #2.

*Section 3.4 I think that the authors did not validate the results of this section using the data. They just calculated the landslide susceptibility based on the scenarios of land use. If the authors want to show one of examples of possibility of LAND-SE, I can understand meaning of this section, but I cannot agree with the last paragraph in this section (L302-306). If the authors want to note the effectiveness of LAND-SE for testing effects of land use change on landslide susceptibility, they have to validate their calculation results.*

The purpose in this section is to describe and analyse the effect of different land use scenarios on landslide susceptibility zonation. In particular, we want to check if the extent of area classified as highly unstable (red area) decrease or increase changing the extent of forest cover. The analysis of the success rate curves (Figure 7) confirms that land use changes affect significantly the spatial distribution of unstable/stable slopes. In particular an increase in the forested areas reduces significantly the extent of slope units classified as highly unstable (red area in fig 7-A1, B1, C1). On the other hand, the decrease of forest coverage (forest fire scenario) increase the percentage of study area classified as highly unstable (red area in fig 7-D1). Moreover, since we have analysed empirical and heuristic synthetic scenarios, their proper validation is possible only when and if such conditions will occur in the study area.

I look forward to hearing your decision soon.

Yours sincerely,